# Spin Trapping of Nitric Oxide by Hemoglobin and Ferrous Diethyldithiocarbamate in Model Tumors Differing in Vascularization

**DOI:** 10.3390/ijms25084172

**Published:** 2024-04-10

**Authors:** Dariusz Szczygieł, Małgorzata Szczygieł, Anna Łaś, Martyna Elas, Roxana Zuziak, Beata K. Płonka, Przemysław M. Płonka

**Affiliations:** 1Department of Biophysics and Cancer Biology, Faculty of Biochemistry, Biophysics and Biotechnology, Jagiellonian University, 31-007 Krakow, Poland; dariusz.szczygiel@uj.edu.pl (D.S.); gosia.szczygiel@uj.edu.pl (M.S.); anna.las@onet.eu (A.Ł.); martyna.elas@uj.edu.pl (M.E.); roxana.zuziak@awf.krakow.pl (R.Z.); beata.plonka@uj.edu.pl (B.K.P.); 2Department of Chemistry and Biochemistry, Institute for Basic Sciences, University of Physical Education, 31-571 Krakow, Poland

**Keywords:** EPR, ESR, hemoglobin, DETC, NO, hemorrhagic necrosis, melanoma, lymphoma, Ehrlich carcinoma

## Abstract

Animal tumors serve as reasonable models for human cancers. Both human and animal tumors often reveal triplet EPR signals of nitrosylhemoglobin (HbNO) as an effect of nitric oxide formation in tumor tissue, where NO is complexed by Hb. In search of factors determining the appearance of nitrosylhemoglobin (HbNO) in solid tumors, we compared the intensities of electron paramagnetic resonance (EPR) signals of various iron–nitrosyl complexes detectable in tumor tissues, in the presence and absence of excess exogenous iron(II) and diethyldithiocarbamate (DETC). Three types of murine tumors, namely, L5178Y lymphoma, amelanotic Cloudman S91 melanoma, and Ehrlich carcinoma (EC) growing in DBA/2 or Swiss mice, were used. The results were analyzed in the context of vascularization determined histochemically using antibodies to CD31. Strong HbNO EPR signals were found in melanoma, i.e., in the tumor with a vast amount of a hemorrhagic necrosis core. Strong Fe(DETC)_2_NO signals could be induced in poorly vascularized EC. In L5178Y, there was a correlation between both types of signals, and in addition, Fe(RS)_2_(NO)_2_ signals of non-heme iron–nitrosyl complexes could be detected. We postulate that HbNO EPR signals appear during active destruction of well-vascularized tumor tissue due to hemorrhagic necrosis. The presence of iron–nitrosyl complexes in tumor tissue is biologically meaningful and defines the evolution of complicated tumor–host interactions.

## 1. Introduction

Nitrosylhemoglobin (HbNO) was the first iron–nitrosyl complex investigated by electron paramagnetic resonance spectroscopy (EPR; alternatively: electron spin resonance—ESR), initially in solutions [1,2], then in tumor tissues [3,4], including human tumor tissues [5,6,7]. It was also found in normal tissues, induced by hypoxia or necrosis [8,9,10,11]. Since the 1960s, HbNO has been employed to study conformational changes of hemoglobin caused by ligand binding [12,13,14]. It has turned out to play a significant role in the investigation of NO generation in tumor and normal tissues due to endotoxemia [15,16], antitumor response [17,18], or allograft rejection [19,20], in particular after the discovery of NO generation by cytotoxic activated macrophages [21].

EPR possesses two features that allow this technique to be used for the measurement of NO in living cells. First: it is a direct process which does not require intermediate phenomena or factors (such as monoclonal antibodies, etc.) to assess NO. Second: it is independent of other necessary procedures like electrophoresis, etc. It is also believed to be the only procedure, apart from polarimetry, to directly measure or record the phenomenon of life.

Even though HbNO was discovered a long time ago (1960s [22,23]), its presence in many animal tumors has not been satisfactorily clarified. It must be emphasized that NO is not only trapped by Hb to produce this complex, but NO itself influences tumor blood supply by vasodilation, stimulation of neovascularization [24], etc., and consequently, the availability of Hb as a spin trap for NO determines the level of HbNO [25]. Hemoglobin is not uniformly distributed in the tumor tissue because it is only present in the blood vessels of the tumor and in areas where blood is perfused into the tumor tissue, being inaccessible in the remaining areas of the tumor.

Another type of endogenous iron–nitrosyl EPR-detectable complex, also called dinitrosyl–iron complex (DNIC), is formed with non-heme iron (Fe(RS)_2_(NO)_2_) [26,27]). Their levels not only depend on the local iron(II) supply to the tissue, but also on the levels of heme targets, and on the local intensity of NO generation [17,18,21,26,28,29,30]. It makes both types of endogenous iron–nitrosyl complexes difficult to use as a direct estimator of NO levels in tumor tissues. Consequently, in EPR-measurements in vivo, exogenous spin traps are usually used, mainly diethyldithiocarbamate (DETC) [31] and its derivatives [32]. Two molecules of the trap initially chelate Fe^2+^ ion (endogenous or exogenous), which eventually coordinates NO to yield the characteristic, EPR-detectable nitrosyl complex Fe(DETC)_2_NO [33], also called the mononitrosyl–iron complex (MNIC) [34]. Even in this case, however, the yield of NO complexing is not the same throughout the tumor. First, generation of NO in solid tissue is heterogeneous [30,31,35]. Second, there are local differences in the conditions necessary for the formation of the complexes, such as the redox environment, accessibility of iron and chelator, solubility of the spin trap, short diffusion distance of NO due to NO scavengers and targets, as well as the elution of the complexes [30,32,35,36,37]. Such situations would lead to the generation of local “hot spots” with high concentrations of trapped NO, often manifested in the EPR spectrum.

Vascularization and blood supply strongly influence tumor growth and invasiveness in vivo, and therefore determine the results of cancer treatment and prompt the important targets for anticancer therapy [38,39,40,41]. Particular types of tumors may differ significantly in their vascularization [42,43,44] and blood content, and this parameter must be taken into careful consideration when designing the appropriate treatment strategy. NO synthesis in tumor tissue not only affects tumor vascularization and blood supply, but also causes tumor growth slowdown, tumor growth arrest, or even tumor regression and necrotization [45,46,47], especially when formed in high quantities [18,24,28].

Necrosis was one of the first factors found to affect HbNO EPR signals in tissues [8,9,11]. Actually, it was induced by such necrotizing agents as concentrated sulfuric acid (pyrolysis [9], or hypoxia [8]), including hypoxia caused by excess nitrite [11]. As necrosis is a common component of solid tumors [48,49], it was suspected that the presence of the primary HbNO signals in intact tumors is caused by non-specific tissue necrosis, or even by hypoxia itself [8,9]. But it has recently become obvious that, in vivo, it was nitrite, not the resulting hypoxia, that is responsible for HbNO elevation [50,51]. However, hypoxia, as a factor prolonging the lifetime of NO, strongly favors HbNO synthesis [36,52]. It is one of the strongest inducers of tumor necrotization [49,53], tumor immunization, and inflammation [48,54,55]. Finally, there are many types of necrosis, both in normal, and tumor tissues, among which ischemic necrosis, typically caused by hypoxia and poor blood supply is usually contrasted with hemorrhagic necrosis [54,55,56,57,58,59]—liquefactive necrosis, in some cases caused by inflammation, and in normal tissues, by disruption of blood vessels and blockage of blood drainage from the organ (e.g., as seen in testicular torsion [56]).

Therefore, the relation between HbNO EPR signals, hypoxia, and necrosis is not as simple as “HbNO EPR signals being the signals of necrosis or of hypoxia”. In tumors, with their ineffective blood flow, leaking vessels and blood extravasation, the amount of blood seems one of the main determinants of the level of HbNO complexes [30,35,60]. The second determinant is the level of NO, resulting primarily from the intensity of cytotoxic activation of tumor cells, mainly macrophages [18,21,61] but also endothelial cells [62,63], or even some tumor cells [24,64,65,66]. The third one, namely the local availability of the high-molecular-weight hemoglobin for NO, may result from the degree of tissue destruction—progress of hemorrhagic necrosis—as normally Hb is expected to be enclosed in circulating erythrocytes, and not in the vicinity of the NO-generating cells [67].

To determine the relation between HbNO and NO in situ, an independent assay for NO is necessary [68,69]. We used iron-DETC, which is the most commonly used NO-metric spin trap, the most hydrophobic one, and the one with the longest retention time in the body [31,32,37]. We are able to easily detect Fe(DETC)_2_NO complexes after injury or during endotoxemia in various organs, including immune privileged organs, like the eye [70] or brain [71,72].

While the dependence of the level of HbNO on all the factors is intricate and cannot be used as a simple indicator of NO level, there is always some resultant HbNO level correlated with a resultant HbNO EPR signal, which can be the primary subject of an independent study. The goal of the present paper is the first step in this direction. We aimed at demonstrating that the level of EPR-detectable HbNO complexes reflects not only gross NO generation, but also important pathological processes in tumor tissue, such as tumor neovascularization, blood extravasation, and various forms of necrosis. To estimate the levels of Fe–NO complexes in tumor tissue independently of Hb, we used iron-DETC as the exogenous spin trap for NO. We also measured the levels of Fe(RS)_2_(NO)_2_ complexes in the investigated tumors. This allowed us to compare the generation of gross NO formation with the level of endogenous Fe–NO complexes in tumors with different vascularization levels, growth rates, immunogenicity, and responses induced by the host. It should elucidate what other factors, in addition to NO levels, affect the levels of Fe–NO complexes, and what information is provided by the particular types of iron–nitrosyl complexes that are formed in tumor tissue.

## 2. Results

### 2.1. Fe–NO Complexes in Tumors Differing in Vascularization

Complexes of nitric oxide with endogenous heme (HbNO) were detected in tumors by EPR spectroscopy. We also recorded complexes of NO and endogenous non-heme iron Fe(RS)_2_(NO)_2_, as well as Fe(DETC)_2_NO complexes with exogenous intraperitoneal (IP) DETC and Fe(II) (endogenous and/or supplemented SC). EPR spectra of paramagnetic Fe–NO adducts in EC, L5178Y and Cloudman S91 tumors are shown in Figure 1.

Apparently, all types of the analyzed nitrosyl complexes were EPR-detectable in the majority of L5178Y tumors. On days 4–5, the signals were low, but steeply increasing afterwards (days 8–11). This tendency was observed in both types of hosts, however, the signals were stronger in the allogenic (Swiss), as compared to the syngeneic host (DBA/2) (Figure 2). All kinds of the analyzed types of Fe–NO complexes were also detected in EC tumors from Swiss mice, where the levels of the adducts rose from day 8 to day 18. However, while the EPR signals of Fe(DETC)_2_NO complexes were very strong, the endogenous adducts Fe(RS)_2_(NO)_2_ and HbNO remained poorly detectable. In all Cloudman S91 tumors growing in DBA/2 mice, only HbNO complexes were detectable, while there were hardly any signals of the non-heme Fe–NO adducts: Fe(RS)_2_(NO)_2_ or Fe(DETC)_2_NO.

Figure 2 presents the quantitative assessment of EPR signals of NO complexes with endogenous hemoglobin and NO complexes with IP exogenously administered DETC and Fe(II) (endogenous and/or supplemented SC) for individual types of tumors differing in vascularization. The highest levels of HbNO complexes were observed in hemorrhagic necrosis of very well-vascularized S91 melanoma tumors and in well-vascularized L5178Y lymphoma tumors growing in the allogeneic Swiss host. Lower levels of HbNO complexes were found in the well-vascularized tissue of L5178Y lymphoma and S91 melanoma tumors growing in the DBA/2 syngeneic host. However, the lowest levels of HbNO complexes were found in poorly vascularized Ehrlich carcinoma tumors. When assessing the levels of Fe(DETC)_2_NO complexes, their highest levels can be seen in poorly vascularized Ehrlich carcinoma tumors and in well-vascularized L5178Y lymphoma tumors in the allogeneic Swiss host. Lower levels of Fe(DETC)_2_NO complexes are found in well-vascularized L5178Y tumors in the DBA/2 syngeneic host, and levels of Fe(DETC)_2_NO complexes present in very well-vascularized S91 melanoma tumors were hardly detected.

Histologic slides were prepared from subcutaneous solid tumors of L5178Y lymphoma, Ehrlich carcinoma, S91 melanoma, and the spleen as the positive control, in which blood vascularization was assessed using immunohistochemical staining for the surface protein CD31, characteristic of the vascular endothelium. Figure 1 shows photographs of stained histological preparations along with gross photographs of the entire tumors and spectra of the representative EPR signals of Fe–NO complexes for a given type of tumor. Table 1 summarizes the quantitative assessment of vascularization parameters in the examined tumors.

MVD and TVA in L5178Y lymphoma tumors depended on the time point of growth: in DBA/2 on day 8 they were about twice as high as on day 5, while the mean size of microvessels was comparable. On day 5, MVD and TVA were slightly higher in Swiss than in DBA/2 mice, but MAM of tumor microvessels was, again, the same. MVD and TVA of EC tumor in Swiss mice were about 10-fold lower than the ones of L5178Y lymphoma; however, the MAM was still similar. In Cloudman S91 melanoma, the mean values of MVD and TVA were about 10-fold higher than in EC. Importantly, MAM values in S91 tumors were about 2–3 times higher than in the other tumors.

Figure 3 reveals that the observed signal of HbNO complexes in various tumor groups is significantly statistically correlated with the total vascular area. The higher the percentage share of blood vessels in the tumor volume, the stronger are the observed HbNO signals in these tumors.

We assessed the correlation of the signal amplitudes between Fe(DETC)_2_NO and HbNO complexes from tumors (Figure 4). This way we were able to obtain information on the amount and availability of NO, which had not been trapped by Hb and, indirectly, on the availability of NO for Hb. In L5178Y lymphoma (observations on day 8 in Swiss host), the signal of HbNO was increasing together with a gradual increase of the Fe(DETC)_2_NO signal up to considerably high levels of NO adducts, whereupon the amplitude of HbNO signals remained constant (about 150 a.u.), although the signal of Fe(DETC)_2_NO adducts could still substantially increase (Figure 4A). A similar shape of the correlation curve between Fe(DETC)_2_NO and HbNO complexes signal in L5178Y tumors was also revealed in the DBA/2 host (observations on day 8), however, the achieved constant level of HbNO signals was lower (about 125 a.u., Figure 4B). A complete absence of correlation between Fe(DETC)_2_NO and HbNO complexes signal amplitudes was observed for EC and Cloudman S91 melanoma tumors. In EC tumors, the signal of HbNO complexes was low (about 30 a.u.) independently of Fe(DETC)_2_NO adducts, their level achieving very high values (Figure 4C). Interestingly, Cloudman S91 melanoma tumors revealed very variable HbNO signals, from very weak in the cortical parts to very strong in the area of central hemorrhagic necrosis, whereas Fe(DETC)_2_NO complexes remained undetectable (Figure 4D).

### 2.2. Fe–NO Complexes in the Areas of Hemorrhagic Necrosis in the Tumors

In the central part of large (above 1 g), very well-vascularized Cloudman S91 melanoma tumors, a distinct core of hemorrhagic necrosis with a variable amount of extravasated blood (as judged by the color of the lesion) was usually observed (see Figure 1), in addition to the cortical layer containing alive cells (fragments of this cortex initiated growth of new tumors after SC inoculation into other animals). In L5178Y tumors, such hemorrhagic necrotic foci were very rare, while in poorly vascularized EC, they were absent, even in advanced tumors.

The EPR signals of the regions of hemorrhagic necrosis were compared with alive cortical layer of murine S91 melanoma and L5178Y lymphoma tumors, and with the peripheral blood of the host tumor. The central hemorrhagic necrosis of advanced S91 and L5178Y tumors revealed strong EPR signals of HbNO, up to 3–4-fold more intense than those of the alive cortical parts of the tumors, while in the peripheral blood they were at the limit of detection (Figure 5A,B). It shows that extensive formation of HbNO takes place after extravasation of blood into the areas of necrosis, but not in blood vessels. Simultaneous diffusion of NO from tumor tissue into blood vessels and the circulating blood is hardly noticeable. S91 tumors did not reveal Fe(DETC)_2_NO signals—they only showed Cu(DETC)_2_ signals confirming the penetration of DETC into the S91 tumor tissue, therefore only HbNO complexes of NO were analyzed here. L5178Y tumors revealed both types of NO complexes, so we showed that in hemorrhagic necrosis, HbNO signals predominated, but in cortical alive tissue, the equilibrium was directed toward Fe(DETC)_2_NO signals, as confirmed by the spectra shown in Figure 5B. In the circulating blood of the tumor host, both HbNO and Fe(DETC)_2_NO signals were hardly detected, despite strong signals in the tumors.

### 2.3. HbNO Complexes in Tumors with Blood Extravasation

Mechanical homogenization of the cortical layer of the tumor tissue results in blood extravasation. This procedure was used because we wanted to check whether blood extravasation in the tumor affected the complexation of NO by Hb. In various experimental tumors, homogenization led to a significant increase of HbNO levels, and for Cloudman S91 melanoma (Figure 6), homogenization always increased HbNO signal amplitude.

## 3. Discussion

In the present paper we analyzed the formation of various Fe–NO complexes in several animal tumors differing in their kinetics of growth, but also in the parameters of blood supply and vascularization. We revealed that the main factors determining the appearance of HbNO signals, in addition to genetic compatibility of tumors and their hosts, are blood supply and the condition of the tissue, its necrosis, and the quality of necrosis.

We showed that the tumors differ in growth. This depends not only on the type of tumor, but also on the type of host, which was reported previously [17,18,30,35,73,74]. The EPR signals of nitrosyl–iron complexes may be associated with the degree of incompatibility between the tumor and the host. The strongest signals are produced in the tumors growing in allogeneic hosts (e.g., L5178Y lymphoma—in Swiss mice) where they grow slowly and reach a small size. Also, the signals of Fe(RS)_2_(NO)_2_ complexes, which become detectable at higher concentrations of NO in the tissue than HbNO [28,30], appear more frequent in such tumors, than when growing in the syngeneic host. Therefore, the described dependence is not only quantitative, but also qualitative.

For the Fe(RS)_2_(NO)_2_ signals, they become detectable only after saturation of heme targets in the tissue [26,28,29,30]. The presence of complexes in the same tissue sample, in addition to HbNO with a strong hyperfine structure, which originates from 5-coordinate signals from α-chains of hemoglobin only partially saturated with NO [12,13], suggests that the local concentration, thus the generation of NO in the tissue, and consequently the efficiency of spin trapping by various kinds of iron, must be heterogeneous [75]. Therefore, there must be local foci of strong NO generation—perhaps associated with local foci of inflammation [28,30,35,60,76]. This is exactly the way some animal tumors grow and undergo necrosis, where local foci of hemorrhagic necrosis appear as a result of strong local antitumor reaction, usually connected with the activity of tumor-infiltrating macrophages [55,77], lymphocytes [76], or granulocytes [54], often in response to therapy [57,76,78].

We found such local regions of Cloudman S91 melanoma tumor tissue with particularly strong HbNO signals to be the necrotic core of the tumor, revealing particularly strong macro blood extravasation, thus of a hemorrhagic type. Also, the alive cortical part revealed strong heterogeneity in the intensity of HbNO EPR signals, indicating at the same time heterogeneity in the NO synthesis. The spots revealing particularly high HbNO signals might represent the initial stages of inflammatory foci which could later on transform into foci of hemorrhagic necrosis (appreciable by routine histology, Figure 1 and Figure 5). In other types of tumors (L5178Y), when foci of hemorrhagic necrosis were found, they also revealed strong HbNO signals. This indicates that not only the amount of NO, which may correspond to the strength of the antitumor reaction of the host [17,18,60,74], is important in observing strong HbNO EPR signals, but also the process of active destruction of the tissue. To confirm this hypothesis we showed that, indeed, such active destruction of alive, vascularized tumor tissue—by homogenization—increased the EPR signal of HbNO (Figure 6). The quantitative aspect of this effect varied between the tumors, but it was impossible to induce the signals by homogenization of normal tissues or tumor tissues which do not reveal HbNO signals while being intact (e.g., Mongolian gerbil melanoma Zeman UJ90, [17,79,80]). Moreover, other methods of inducing NO synthesis, namely ischemia followed by homogenization of heart muscle, resulted in a similar effect [81]. Finally, the reaction of isolated lung to LPS based on the blockage of lung blood circulation and local blood extravasation also lead to the induction of HbNO in the tissue [16].

Interestingly, in the isolated lung setup [16], no HbNO signal could be detected in the blood flowing out from the organ, even after reducing the whole material with excess sodium dithionite. HbNO complexes were formed and detected only locally, at the site of the most severe tissue injury. A similar situation was observed with our findings: only very weak HbNO signals (if any) could be detected in the circulating peripheral blood of the tumor-carrying animals, with the absence of other types of signals, while both HbNO and Fe(DETC)_2_NO could easily be detected in endotoxic shock in the blood [15,70]. It indicates that the tumor tissue signals are produced locally in the tumors [31,32], mostly in the extravascular space.

All these findings equivocally demonstrate that a crucial factor that determines the detection of EPR HbNO signals in the tissue is cancellation of barriers between the sites where NO is formed (cytoplasm of cells responsible for inflammation, mainly macrophages, perhaps also endothelium and neutrophils) and where it is trapped by Hb (initially inside of erythrocytes in tumor capillaries). One mechanism involves local inflammation leading to hemorrhagic necrosis, while another one is tissue homogenization. Perhaps, however, the actual mechanism is uniform: the recently described special type of cell death—the “non-apoptotic” caspase-1-driven pyroptosis—leads to physical perforation of cell membranes [82,83,84]. This phenomenon mainly (but not solely, [82]) signifies activated macrophages, causing discontinuity of their plasma membrane [84] and intensifying the progress of inflammation in the tissue [82,83,84].

One more factor, however, should be considered here, namely, the quality and intensity of tumor blood supply. If the supply is not satisfactory, the HbNO complexes will be formed with difficulty even in the situation of strong antitumor reaction of the host. We showed that the type of Fe–NO complexes detectable in L5178Y lymphoma depends on the color of the tissue (in pale tissues with poor vascularization non-heme-iron complexes predominated, and in red parts with good blood supply mainly HbNO was detectable, [35]), which was further confirmed in detail [30,60]. Here, we show that, indeed, the origin of particular EPR samples can be easily determined with the naked eye just by the macroscopic coloration of the icicles (Figure 1C). And, more importantly, while tumor vascularization and blood supply differ in various parameters, their extreme values are associated with the appearance of extreme values of HbNO-type complexes, namely Cloudman S91 melanoma (in particular hemorrhagic necrosis) and Ehrlich cancer (the poorest vascularization and blood supply).

The values obtained for L5178Y lymphoma are additionally interfered with, not only by the type of the host, but also by the relative duration of growth of particular tumors, and the type of necrosis. An additional independent parameter determining NO generation in the tissue would, therefore, be necessary to interpret properly quantitative changes in HbNO EPR signals during tumor growth. This was the reason why we additionally checked the formation of nitrosyl–iron complexes in the presence of excess exogenous chelator DETC, and exogenous iron. It was convenient because the analytical features of HbNO and Fe(DETC)_2_NO EPR signals do not overlap (see Section 4.7). Initially we checked the effects of exogenous iron, which turned out to improve partially the detectability of Fe(DETC)_2_NO complexes in EC, but not in L5178Y tumors. This is consistent with previous observations made by us and other authors about the ability to accumulate iron by L5178Y lymphoma [60,85]. Consequently, it is logical to expect that the levels of endogenous non-heme iron–dinitrosyl complexes Fe(RS)_2_(NO)_2_ must be lower in EC tumors than in L5178Y lymphoma, which has been confirmed here. In the case of Cloudman S91 melanoma in DBA/2, we were unable to detect any Fe(DETC)_2_NO signals even in the excess of exogenous iron, both in the alive cortex and in the necrotic core of the tumors. Only some tumors growing very slowly in allogeneic hosts (“pseudo-DBA/2” mice—DBA/2 mice crossed for more than 30 generations in an outbred system in our animal facility) revealed considerable Fe(RS)_2_(NO)_2_ signals.

Our attempts to correlate the intensities of HbNO with Fe(DETC)_2_NO signals gave a clear picture of the dynamics of the formation of various types of nitrosyl iron complexes in animal tumors. In the tumors with the highest amount of hemorrhagic necrosis and the highest parameters of blood supply (Cloudman S91 melanoma), HbNO did not correlate with Fe(DETC)_2_NO. Even the addition of DETC and iron(II) did not result in the formation of these complexes, nor did it affect the intensity of HbNO complexes, as though the whole accessible NO had been trapped beforehand by the abundant Hb, in the close vicinity of the sites of NO synthesis. The presence of Cu(DETC)_2_ complexes [31,86,87], both in the cortical layer and even in the central necrosis, provides evidence that DETC could even diffuse into large tumors.

The Ehrlich carcinoma tumors [88] revealed low intensity of HbNO and Fe(RS)_2_(NO)_2_ complexes, and poor blood supply. The addition of DETC and iron resulted here in the formation of relatively strong Fe(DETC)_2_NO complexes, however, without any influence on HbNO signals, which remained low, as though there was a pool of NO formed in the tissue that was not accessible for trapping by Hb. Meanwhile in L5178Y lymphoma, a correlation between HbNO and Fe(DETC)_2_NO complexes could be observed (Figure 4), and a plateau was created at extremely high intensities of HbNO signals, while Fe(DETC)_2_NO signals still increased, as if the non-heme iron centers were not saturated (like the curve for EC, but moved upward). The value of the plateau was higher in allogeneic (Swiss) mice and was expected to react stronger to the presence of the tumor than in syngeneic (DBA/2) hosts. The parameters describing blood supply were here only indirectly correlated with signal intensities, depending also on the day of tumor growth (for young tumors the tumor vascularization is expected to be poorer than in old tumors), and on the type of the host.

Melanoma [89] and its metastases [90,91] often (more often than carcinomas, however less often than leukemias) undergo spontaneous regression [92,93] with the involvement of immune mechanisms [94,95,96,97,98,99], in which hemorrhagic necrosis has been found to be typically responsible for this regression [94,95,96,97,98,99]. Even the primary tumors which were the source of transplantable lines of Cloudman S91 [100,101] and B16 melanoma [102,103] were described as hemorrhagic. This is consistent with our results, and also explains some underlying mechanisms—the most prominent development of hemorrhagic necrosis in the Cloudman S91 tissue with particularly large diameters of blood vessels.

Thomsen et al. (1992) [55] suggested that it is the inflammatory effector factors released by activated macrophages that actually cause hemorrhagic necrosis. Moreover, since the late 1920s it has been known [104,105,106,107,108,109,110] that hemorrhagic necrosis can be induced in normal tissues or in tumors by non-specific activation of the immune system. Therefore, it is not hemorrhagic necrosis that is the actual primary cause of a high HbNO level, but NO itself. It is one of the important inducers of hemorrhagic necrosis, which itself thereafter increases the possibility to be trapped by Hb, and in the positive feedback (as every necrosis stimulating inflammation [48]), it intensifies the generation of NO even more. Indeed, unlike apoptosis, pyroptosis aims at facilitating inflammation by the release of proinflammatory intracellular contents [83,84], including IL-1 and IL-6 [82], in response to various pathological stimuli, mainly microbes [83,84] but also cancer [82,83]. For this reason, strongly necrotizing tumors not always undergo regression, on the contrary, inflammation and necrotization are sometimes poor prognostic factors [54], which may be to some extent be explained by the toxicity of extravasated hemoglobin, and paradoxically, by the anti-heme activity of NO itself [111].

All these results can be interpreted in the following way: DETC, always administered shortly before the EPR measurement, easily penetrates cellular membranes and forms insoluble complexes with Fe(II) in situ, trapping all or almost all the available NO (as was the case with EC). But the affinity of the heme iron for NO is much higher than that of the nonheme iron(II), and initially, all the available heme traps—mainly the ones originating from the extravasated blood—become saturated with NO (Cloudman S91). This is a long process, but HbNO complexes are quite stable and accumulate in the tissue. The excess of DETC and iron traps the NO not otherwise accessible for Hb (L5178Y). If the blood supply is satisfactory and extravasation (formation of foci of hemorrhagic necrosis) intensive, almost all the NO is trapped by Hb (Cloudman S91 melanoma) beforehand. The amount of the necrotic “core” in these tumors is so big that the distribution of insoluble Fe(DETC)_2_ is probably limited compared to the cortex.

The absence of any of the signals in the peripheral blood, but its increase in homogenized tissue, additionally supports the hypothesis of the local formation and local trapping of NO by hemoglobin stored in the extravascular space of the tissue rather than circulating in the blood. While this supports the notion of strong heterogeneity of NO activity within the tumor [75], the tumor HbNO signal may still be treated as the signal of a vascularized tissue undergoing destruction, with Fe(DETC)_2_NO as the estimator of the formation of NO in a non-necrotizing tumor. The signal of Fe(RS)_2_(NO)_2_ partially reflects the availability of iron(II) in the tumor tissue. It may also represent the reducible character of the local environment (as in oxidative environments, Fe(II) easily undergoes oxidation to Fe(III)), and the extreme NO generation, as the affinity of low-molecular weight endogenous Fe(II) complexes for NO is low compared to Hb.

## 4. Materials and Methods

### 4.1. Chemicals

Phosphate-buffered saline (PBS) was obtained from BIOMED Wytwornia Surowic i Szczepionek (Lublin, Poland) and ICN Biomedicals Inc. (Aurora, OH, USA); ferrous sulfate (FeSO_4_·7H_2_O), sodium citrate and sodium diethyldithiocarbamate (DETC) were obtained from Sigma-Aldrich (St. Louis, MO, USA); acetone, eosin, and formalin were manufactured by Polskie Odczynniki Chemiczne (Gliwice, Poland); paraffin for embedding came from Histoplast, Thermo Shandon (Runcorn, UK). Protein block serum-free, Biotin Blocking System, Avidin Solution and Biotin Solution were from Dako North America Inc (Carpinteria, CA, USA); hematoxylin (Mayer hematoxylin) was from Stamar (Dąbrowa Górnicza, Poland), and 1o anti-mouse CD31 (PECAM-1) from Pharmingen, Becton Dickinson Bioscences (Warsaw, Poland). Biotinylated anti-rat IgG (H+L), Vectastein ABC and DAB substrate kit for peroxidase were from Vector Laboratories Inc (Burlingame, CA, USA), pentobarbital, xylazine and ketamine from Biowet (Puławy, Poland).

### 4.2. Animals and Tumor Cell Lines

The experimental animals (male and female 8–10 weeks old mice) were purchased from the Animal Breeding Facility of the Institute of Pediatric, Collegium Medicum, Jagiellonian University, Krakow, Poland. They were kept with permanent access to standard rodent chow and water under standard conditions, in community cages and a 12-h day/night regime.

Murine lymphoma L5178Y (L5178Y-R sub-line, [112,113,114]) was maintained in vivo in inbred DBA/2—the natural syngeneic host [114] for this tumor line and outbred Swiss mice as the allogeneic host [17,30,35,60,73]. Solid L5178Y tumors were inoculated SC using 2 × 10^7^ ascitic cells or 30 mg of very small solid tumor fragments. Solid tumors of murine Ehrlich carcinoma (EC) were inoculated SC in outbred Swiss mice (the natural host [115]) using 2 × 10^7^ ascitic cells. Murine Cloudman S91 melanoma (amelanotic form [100]) was inoculated SC in the natural syngeneic DBA/2 host [17,68,90] using 30 mg of very small solid tumor fragments or 1 × 10^6^ cells cultured in vitro. As the SC growth of the tumors used is anisotropic, particularly in the case of EC, the mean diameter d of the tumor growing in vivo was calculated as the geometrical mean, according to Schreck [116]: d = (a × b × c)^1/3^, where a, b and c are the perpendicular tumor diameters. At scheduled time points, the animals were euthanized (overdose of ketamine and xylazine) and the tumors were extracted. The selection of the tumor growth day on which they were to be evaluated for the levels of NO complexes and vascularization parameters was based on their growth kinetics and the predetermined time point of attainment of a diameter of about 1 cm. L5178Y lymphoma tumors were the fastest growing and reached a diameter of about 1 cm on day 8. Ehrlich carcinoma tumors grew slower and attained a diameter of 1 cm around day 18. The slowest are Cloudman S91 tumors and these were examined on day 27, when they had reached a diameter of more than 1 cm and revealed central hemorrhagic necrosis. Authorizations to carry out the experiments were granted by the 1st Local Committee for Animal Research in Kraków (opinions No. 15/OP/2002, 18/OP/2003, 25/2009 and 14/2015).

Solid tumors in vivo, and NO spin trapping with DETC: Some of the tumors were used to spin-trap NO using exogenous DETC. The tumor-carrying animals were injected IP with the exogenous spin trap: sodium diethyldithiocarbamate (500 mg/kg), 30 min before tumor extraction [86]. Some of the DETC-treated animals were also co-treated with excess exogenous iron (SC, FeSO_4_·7H_2_O, 50 mg/kg, in 250 mg/kg sodium citrate in the tumor region) in parallel to IP DETC injection [117]. To remove oxygen, all the reagents were dissolved in PBS bubbled with gaseous nitrogen for at least 30 min before use.

### 4.3. EPR Sample Preparation

After excision, tumor tissue was weighed, photographed, the fraction of cortical alive tissue was separated from the central necrotic core (if present), and cut into smaller fragments, so as to be subsequently packed in standard glass tubes and immediately frozen in liquid nitrogen. The samples (icicles, ca., 4 mm in diameter and 2 cm in length) were stored at −80 °C until EPR measurement (no longer than 3 months).

### 4.4. Analysis of Tumor Necrosis

For the purposes of this paper, it was enough to distinguish between the type of necrosis: hemorrhagic versus ischemic, the former being a type of liquefactive, and the latter of coagulative necrosis. The necrosis was classified as red, hemorrhagic if it possessed a liquid consistency and red coloration, otherwise classified as “non-hemorrhagic”. The common method of quantification of red (hemorrhagic) necrosis [55,77,118] is also based on the arbitrary estimation of some fragments of histological sections, but a tumor used for histology cannot be measured by EPR in search of NO complexes, and consequently, the “yes or no” differentiation between the two types of necrosis [57] seemed justified in our case. Additionally, the degree of necrotization of selected tumors and EPR samples was documented photographically, while other parts of the tumors were fixed in 5% buffered formalin (pH 7.4) and checked using routine histology (hematoxylin and eosin staining, H&E [119]).

### 4.5. Tumor Homogenization

Some parts of the alive cortical fraction of solid tumors were placed in a glass Potter homogenizer, and homogenized manually, whereupon standard EPR homogenate samples were prepared and stored just like solid tumor samples. The EPR signals were subsequently compared with the signals of the non-homogenized parts of the corresponding intact tumors.

### 4.6. Blood EPR Analysis

Whole blood of tumor-carrying or control untreated animals was taken from the left ventricle of the heart with a plastic disposable syringe, placed in the EPR tube (0.5 mL) used to prepare tumor samples, and immediately frozen in liquid nitrogen [16,70].

### 4.7. EPR Measurement

The icicles were pushed into a quartz Dewar flask and measured at 77 K, using a Varian E-3 spectrometer with a rectangular TE 102 cavity. EPR spectra were recorded at X-band (9.15 GHz), at 3280 ± 250 Gs field center and sweep, 4 mW microwave power, 0.3 s time constant, 240 s acquisition time, 10 Gs modulation amplitude, and 20,000–400,000 receiver gain. The spectra were recorded in a digital form in triplicate and averaged. In all cases, the EPR signal amplitudes were normalized according to the constant receiver gain (200,000), and the constant mass of tissue (400 mg), on the basis of a calibration curve prepared for the particular resonant cavity and the applied sample geometry. The intensity of Fe(DETC)_2_NO signals was expressed as the amplitude of the signal at g_⊥_ = 2.035, and that of HbNO as the amplitude of the first, or (measurements of HbNO subsequent to Fe(DETC)_2_NO in the same sample) the third constituent of the hyperfine splitting. Intensity of Fe(DETC)_2_NO signals in tumors was estimated as the amplitude of the third (high-field [28,70,71,72,86,87]) constituent of the hyperfine structure. Particular examples of quantification of the spectra [25,34] are depicted in Figure 7. Simulations of EPR spectra of various types of HbNO complexes are presented in the cited paper: Dutka et al., 2019 (a chapter) [25].

### 4.8. Tumor Microvascularization Analysis

Tumor blood vessels were stained immunohistochemically using CD31 as the endothelium marker by the avidin–biotin–peroxidase detection. Tissue sections were treated with hematoxylin to enhance contrast between protein and the background. The negative control was prepared accordingly, without anti-CD31 antibodies. The stained slides were scanned using a light microscope, Nikon Eclipse TS 100, and a video camera, Sony Exwave, model No. SSC-DC54AP. At a magnification of 40×, three foci of dense vascular staining were chosen. Then photographs were taken using a magnification of 100×. Images were then digitalized, filtered, and enhanced. Non-specific staining areas of a surface of less than 40 pixels (smaller than a single cell) were removed. For the analysis, the freeware program ImageJ (version 1.30) was used (http://rsbweb.nih.gov/ij/).

Vascular density was quantified according to Weidner [121,122] as the mean number of microvessels per mm^2^ (MVD) and as total microvessel area (TVA). Size of tumor microvessels was measured as the mean area of single microvessel (MAM = TVA/MVD).

### 4.9. Statistics

All the results are presented as means of values ± SD or ± SE. Statistical significance was determined by two-tailed, independent Student’s-*t* test. In the case of tumor homogenates, to establish the statistical significance of the increase in the HbNO signal due to homogenization, in relation to the original solid tumor, paired Student’s-*t* test was used. The statistical significance of differences between the variances was tested by the Snedecor F-test. A difference was accepted as significant for *p* < 0.05.

## 5. Conclusions

Various animal tumors differ in the presence and intensity of HbNO and Fe(DETC)_2_NO signals. The formation of EPR-detectable HbNO complexes in tumor tissues in vivo results from three independent factors: (i) local NO concentration resulting mainly from the strength of the host antitumor response and induction of NO synthesis in tumor in vivo, (ii) tumor blood supply resulting from parameters of tumor vascularization, and (iii) the accessibility of spin traps in the vicinity of NO generation sites resulting from blood extravasation and tissue destruction leading to the formation of foci of hemorrhagic necrosis, which are the main spots of HbNO accumulation in solid tumors. EPR as a method of estimating NO complexes in tumor tissues should be performed using both exogenous and endogenous spin traps. In this way, we obtain a complete picture of NO formation, which may be disturbed due to the heterogeneous structure of the tumor/tumor vasculature/areas of necrosis/areas of extravasation. While its detailed analysis may be difficult, the message conveyed by the presence and intensity of the HbNO signal (and other types of iron–nitrosyl complexes) is valuable and has biological meaning, characterizing interactions between tumors and their hosts.

## Figures and Tables

**Figure 1 ijms-25-04172-f001:**
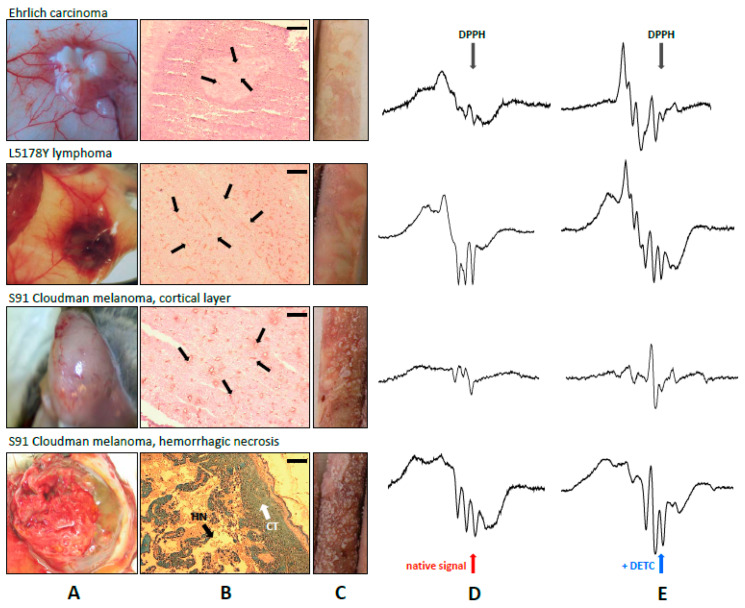
Gross and histochemical differences in vascularization and blood supply in various tumors. From top to bottom: Ehrlich carcinoma, L5178Y lymphoma, S91 Cloudman melanoma. Gross appearance during the autopsy (**A**); CD31 expression as a marker of endothelium marked by thin black arrows. Scale bar 0.2 mm (**B**). In S91 melanoma, a large area of hemorrhagic necrosis was seen (H&E staining; thick black arrowheads, HN), in addition to the alive cortical tissue (thick white arrowheads, CT). The differences in blood supply may be noted when comparing the color of the frozen EPR samples (**C**). Records of EPR spectra of NO complexes in tumors without (**D**), or with exogenous diethyldithiocarbamate (DETC) (**E**). Various proportions of HbNO and Fe(DETC)_2_NO complexes, and in Cloudman S91 melanoma, a strong dependence on the region of solid tumor (cortical tissue or central hemorrhagic necrosis) were seen. Note the presence of Cu(DETC)_2_ tetraplet signal in the tumors of Cloudman S91 melanoma. EPR parameters—see Section 4.

**Figure 2 ijms-25-04172-f002:**
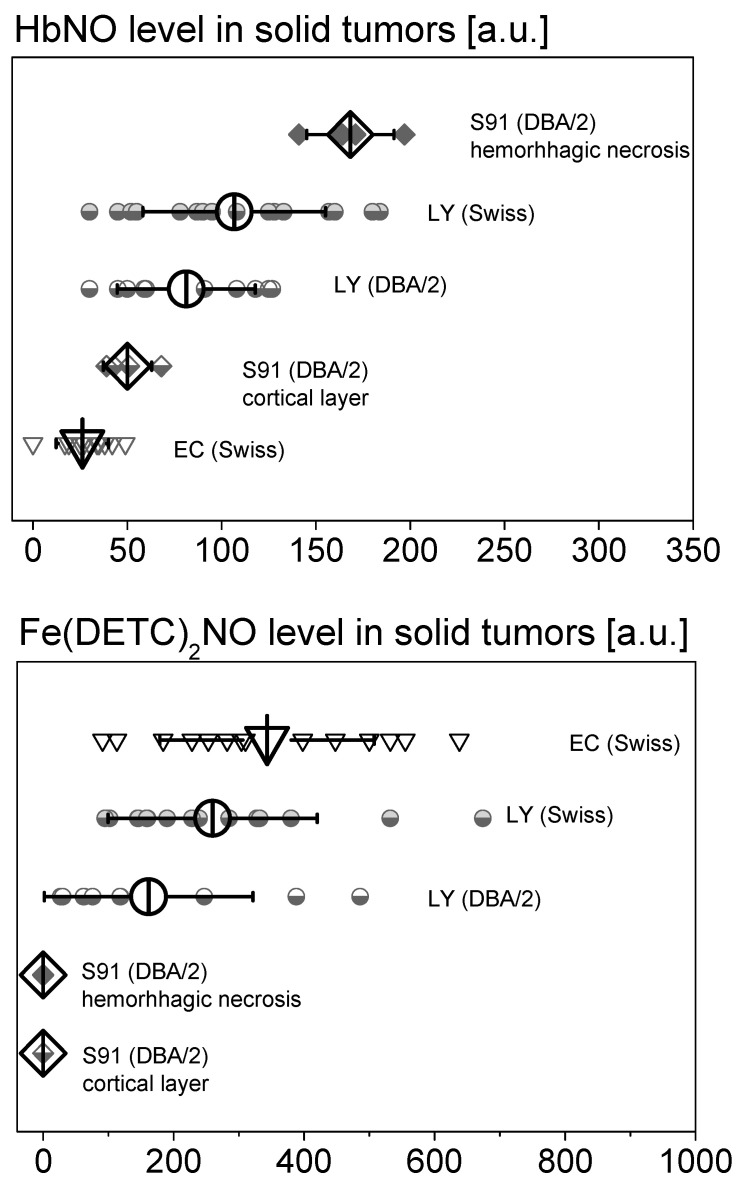
The influence of tumor type on the amplitudes of EPR signals of HbNO and Fe(DETC)_2_NO complexes in subcutaneous solid tumors in mice with administered DETC. L5178Y lymphoma tumors (LY) on day 8, Ehrlich carcinoma tumors (EC) on days 13–18, and Cloudman melanoma tumors (S91) on day 27 in DBA/2 or Swiss mice. Note the strong dependence of the intensity of HbNO signal on the region of Cloudman S91 melanoma tumors and the lack of Fe(DETC)_2_NO complexes in these tumors. Means of N independent tumors (indicated over each bar) ± SD. Parameters of measurement—see Section 4.

**Figure 3 ijms-25-04172-f003:**
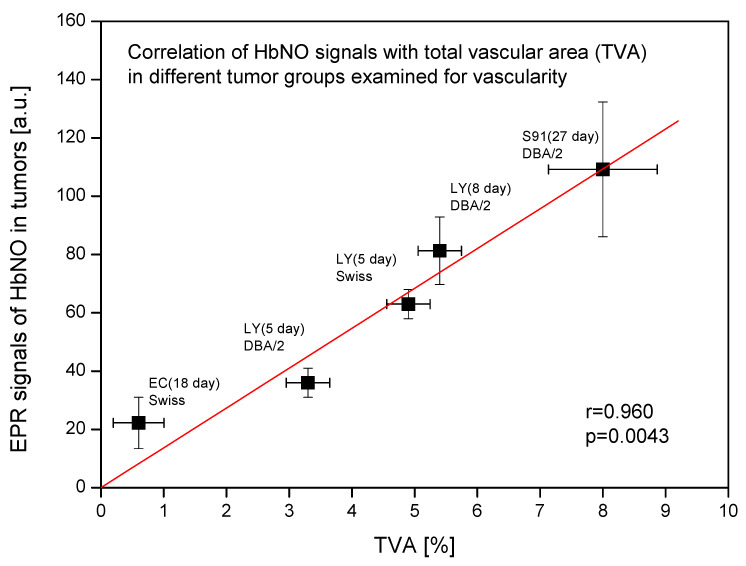
Correlation of EPR signals of HbNO complexes with total vascular area (TVA) in different tumor groups examined for vascular parameters. Amplitudes of EPR signals of HbNO complexes and TVA in subcutaneous solid tumors growing in mice. Ehrlich carcinoma (EC) tumors on day 18 in Swiss mice; L5178Y lymphoma (LY) tumors on days 5 and 8 in DBA/2 or Swiss mice; Cloudman melanoma (S91) tumors on day 27 in DBA/2 mice (mean HbNO signal from both necrotic and non-necrotic S91 tumor areas). Data presented as means ± SE. Linear correlation coefficient r and statistical significance *p* of the correlation were calculated.

**Figure 4 ijms-25-04172-f004:**
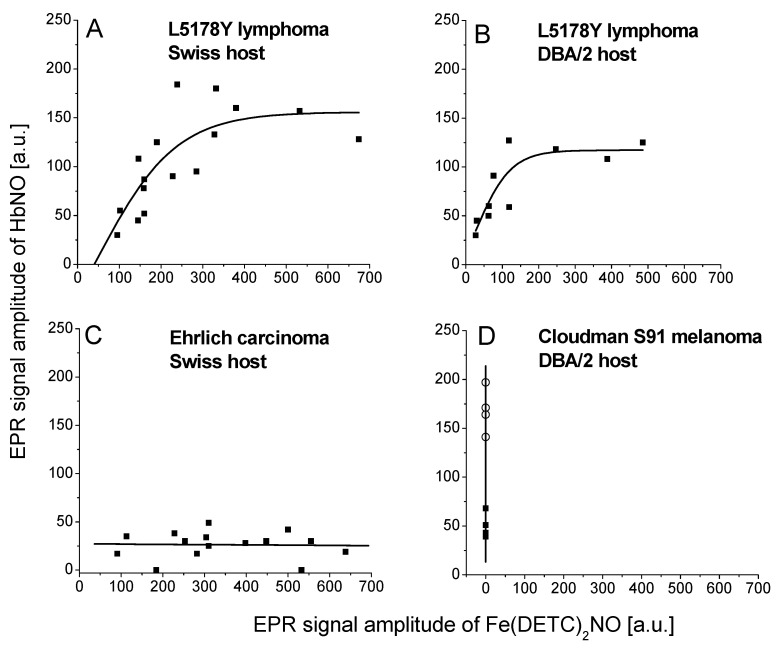
Correlation between EPR signal amplitudes of Fe(DETC)_2_NO (amplitude of 3rd hyperfine line), and HbNO complexes (amplitude of 3rd hyperfine line) in solid tumors. Parameters of measurement—see Section 4. The data were pooled from all the experiments with the use of exogenous DETC for L5178Y lymphoma in Swiss host (**A**), L5178Y lymphoma in DBA/2 host (**B**), Ehrlich carcinoma in Swiss host (**C**), and Cloudman S91 melanoma (cortical layer [■] and hemorrhagic necrosis [◯]), growing in DBA/2 host (**D**).

**Figure 5 ijms-25-04172-f005:**
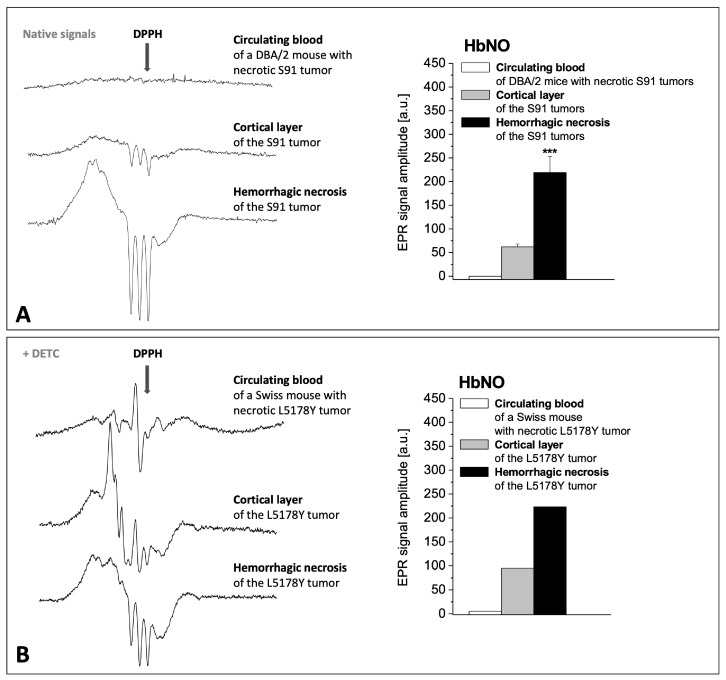
The influence of hemorrhagic necrosis development in tumor tissue on the signals of HbNO complexes in the tumor and in the peripheral blood of the host. Representative EPR spectra of NO complexes (left side) and amplitude quantification of HbNO signals (right side, mean ± SE, *** *p* < 0.001) in tumor and in circulating blood of their host. (**A**) Melanoma S91 growing in DBA/2 host (without DETC injection; N = 15). (**B**) Lymphoma L5178Y growing in Swiss host (injected with DETC; a rare single case of L5178Y tumor with central hemorrhagic necrosis). (I) Circulating blood of the tumor host, (II) cortical layer of the tumor, (III) hemorrhagic necrosis of the tumor.

**Figure 6 ijms-25-04172-f006:**
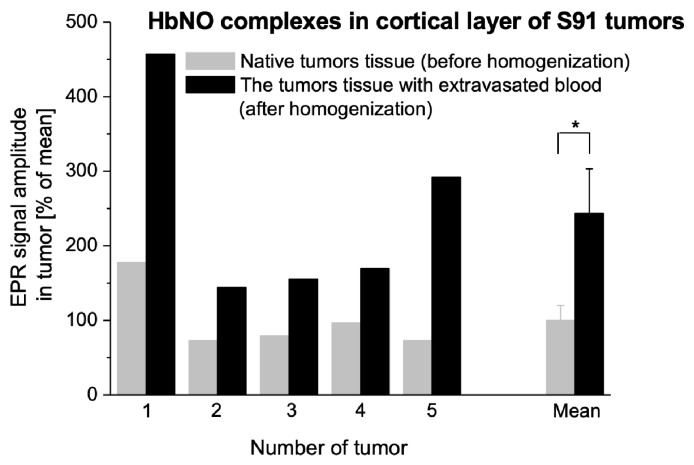
The influence of blood extravasation into tumor tissue caused by mechanical destruction of vessels on the formation of HbNO complexes. HbNO signal in solid tumors of Cloudman S91 melanoma growing SC in DBA/2 hosts; increase of the 2nd hyperfine line of HbNO signal due to tumor tissue homogenization. 1–5: values obtained for 5 independent tumors, mean: comparison of the means obtained for tumors 1–5 ± SE. EPR measurements according to Section 4 (* *p* < 0.05 by paired Student’s-*t* test).

**Figure 7 ijms-25-04172-f007:**
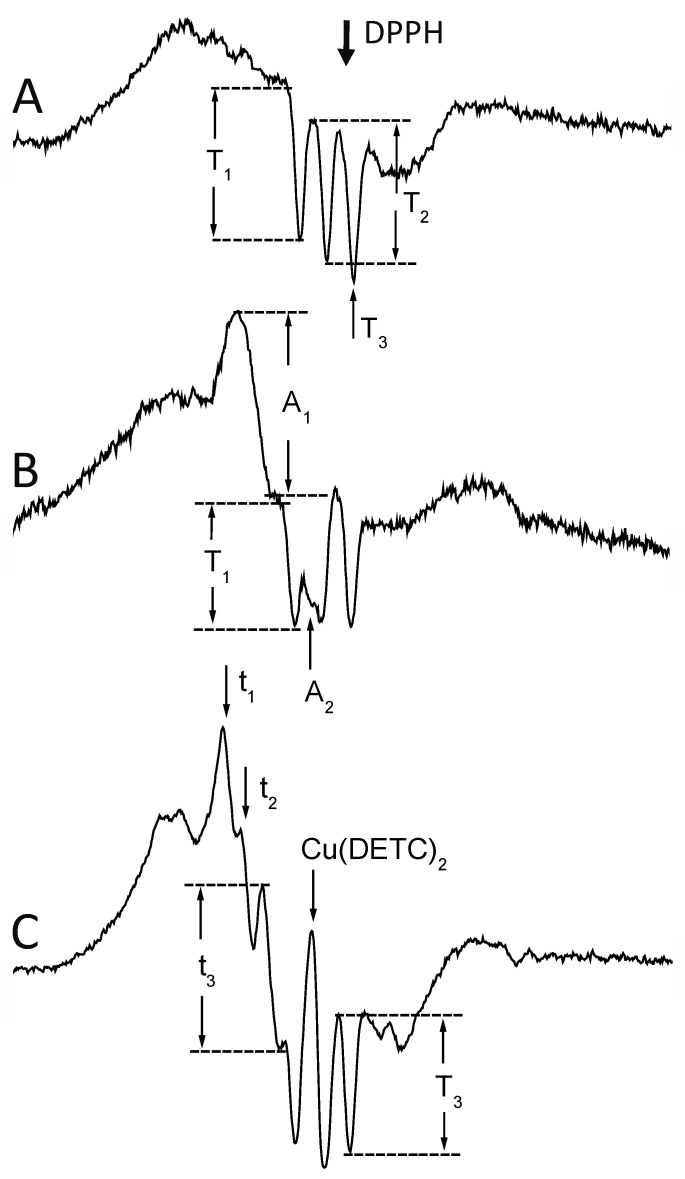
Typical EPR signals of iron–nitrosyl complexes detectable in the investigated animal tumors. (**A**) Native melanoma S91. HbNO signal (g = 2.012, A^N^ = 1.7 mT [1,2,13,14]) of 5-coordinate alpha1 nitrosylated chains revealing hyperfine splitting, and their first (T_1_), and second (T_2_) amplitude. The 3rd component (T_3_) overlaps with the free radical signal (DPPH, g = 2.0037 [120]). (**B**) Native lymphoma L5178Y. Quantitation of the non-heme iron–nitrosyl complex signal (g_⊥_ = 2.035, g_II_ = 2.02 [26,27]) in the presence of HbNO signals. The amplitude of the feature at g_⊥_ (A_1_) was used, as at g_II,_ the signal (A_2_) partially overlaps with the 2nd component of the HbNO hyperfine. T_1_ = the 1st hyperfine of HbNO. (**C**) Lymphoma L5178Y with exogenous DETC. In the presence of the DETC, due to the Cu(DETC)_2_ tetraplet signal (g = 2.024, A^Cu^ = 4.9 Gs [31,86,87]; see Figure 1), only the t_3_ amplitude of the 3rd (high field) component of the Fe(DETC)_2_NO hyperfine (g_⊥_ = 2.035, g_II_ = 2.02, A^N^ = 1.3 mT [31,33,86]) could be used, and the 3rd (high-field) component of the HbNO triplet. t_1_, t_2_ = amplitudes of 1st and 2nd component of Fe(DETC)_2_NO. For parameters of measurement—see Section 4.

**Table 1 ijms-25-04172-t001:** Parameters of solid tumor vascularization in L5178Y, EC, and S91 tumors. MVD = microvessel density (number of vessels/mm^2^); TVA = total vascular area (percentage of area occupied by vessels); MAM = mean area of microvessels (vessels size (μm^2^) calculated as TVA/MVD). The means were obtained from 3 histological sections (L5178Y lymphoma, Ehrlich carcinoma, spleen) and 9 histological sections (Cloudman S91 melanoma). All the results are expressed as mean ± SD.

Tumor Growth Parameters			Tumor Vascularization Parameters		
Solid Tumors	Host	Time [Days]	TVA	MVD	MAM
Ehrlich carcinoma	Swiss	18	0.6 ± 0.7	23 ± 29	230 ± 121
L5178Y lymphoma	DBA/2	5	3.3 ± 0.6	185 ± 25	176 ± 71
	Swiss	5	4.9 ± 0.6	265 ± 11	190 ± 112
	DBA/2	8	5.4 ± 0.6	354 ± 26	150 ± 97
S91 melanoma	DBA/2	27	8.0 ± 2.6	199 ± 73	490 ± 143
Spleen (control)	DBA/2	-	2.8 ± 0.1	841 ± 120	40 ± 18

## Data Availability

Data is available upon reasonable request.

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
