# Peer review of "Spin Trapping of Nitric Oxide by Hemoglobin and Ferrous Diethyldithiocarbamate in Model Tumors Differing in Vascularization"

_ijms, 2024, doi:10.3390/ijms25084172_

Round 1

Reviewer 1 Report

Comments and Suggestions for Authors

The manuscript "Spin trapping of nitric oxide by hemoglobin and ferrous diethyldithiocarbamate in model tumors differing in vascularization" by authors D. Szczygieł, M. Szczygieł, A. Łaś, M. Elas, R. Zuziak, B.K. Płonka, and P.M. Płonka is devoted to determination of iron-nitrosyl complexes in tumor tissues originated from various types of cancer. Dependence of NO production and content of HbNO and iron-nitrosyl complex Fe(DETC)2NO (reflecting NO not bound to Hb) on tumor vascularization, its structure and necrotic changes has been studied in detail.

Despite the complicity and of study described in the manuscript, the practical prospects of the work remain unclear. If analysis of NO content within tumor or healthy tissues by EPR spectrometry or other technique have some diagnostic value, it must be emphasized in the text. If the study was aimed just to estimate the ratio between HbNO and free NO, the practical reasons to do it has to be outlined as well.

I recommend to provide EPR spectra corresponding to different tumor samples as well as images mentioned in Section "Materials and Methods" in Supplementary information. Also, the origin of the samples which EPR  spectra are presented at Figure 6 should be specified.

I believe that manuscript by D. Szczygiel and co-authors can be published in Special Issue "Nitric Oxide Biosynthesis Pathway and Nitric Oxide Signaling in Human Diseases" of IJMS after appropriate improvement: practical value of the method described or perspectives of its practical application must be discussed in the text; title of Fig. 6 should be clarified; Supplementary materials (images, EPR spectra) should be provided.

Comments on the Quality of English Language

Line 54: "Exogenous" should be instead of "egzogenous"

Line 436: "Biotin" should be instead of "Biotyn"

Author Response

The manuscript "Spin trapping of nitric oxide by hemoglobin and ferrous diethyldithiocarbamate in model tumors differing in vascularization" by authors D. Szczygieł, M. Szczygieł, A. Łaś, M. Elas, R. Zuziak, B.K. Płonka, and P.M. Płonka is devoted to determination of iron-nitrosyl complexes in tumor tissues originated from various types of cancer. Dependence of NO production and content of HbNO and iron-nitrosyl complex Fe(DETC)2NO (reflecting NO not bound to Hb) on tumor vascularization, its structure and necrotic changes has been studied in detail.

Despite the complicity and of study described in the manuscript, the practical prospects of the work remain unclear. If analysis of NO content within tumor or healthy tissues by EPR spectrometry or other technique have some diagnostic value, it must be emphasized in the text.

EPR measurements of various tumorsreveal the frequent occurrence of a triplet free radical signal originating from 5-coordinate iron(II) alpha1 nitrosylated chains of hemoglobin (HbNO). In this work, we showed how the presence of HbNO signals depends on various parameters of tumor tissue structure: 1) development of tumor tissue vascularization, 2) blood extravasation, 3) formation of areas of hemorrhagic necrosis in well-vascularized tumor tissue. The results allow us to conclude that strong HbNO signals in tumors occur only when the tumor is well vascularized, and at the same time, there is a process of extravasation of blood to the areas of tumor tissue, because most of the NO produced in tumors is located outside its blood vessels (very weak HbNO signals in peripheral blood and strong HbNO signals in tumor tissue). Areas of hemorrhagic necrosis are such places in the tumor that are characterized by very strong HbNO signals, because they occur in well-perfused tumors, there is an intensive process of extravasation of blood into the necrotic tumor tissue, and in these necrotic areas there is intensive production of NO, which allows for efficient formation of HbNO complexes.

We postulate that the presence of strong HbNO signals in tumors has a diagnostic value, which tells us that such tumors are well supplied with blood; in some areas of the tumor tissue, tissue destruction occurs and blood extravasates into these areas, and, consequently, the formation of foci of hemorrhagic necrosis. It is known that a high degree of vascularization and the presence of hemorrhagic necrosis are risk factors in various types of cancer and correlate with the long-term course of cancer, e.g. due to the increased possibility of metastasis formation inwell vascularized tumors, difficult penetration of drugs into necrotic areas due to the lack of vascularization, or lower effectiveness of radiotherapy in necrotic areas due to the hypoxia.

If the study was aimed just to estimate the ratio between HbNO and free NO, the practical reasons to do it has to be outlined as well.

In this study, the level of NO unbound to hemoglobin was assessed using an exogenous Fe(DETC)2 trap. The measurement of both types of HbNO and Fe(DETC)2NO complexes in tumors allows for a much better assessment of NO production than when using only an endogenous or only an exogenous trap. NO is trapped mainly by Hb in areas of strong blood extravasation into the tissue in well-vascularized and hemorrhagic tumors, while NO is trapped mainly by Fe(DETC)2 in areas where Hb is difficult to access, i.e. in poorly vascularized tumors or in well-vascularized tumors, but outside the areas of hemorrhagic necrosis, or circulating in leak-proof vessels. The HbNO/Fe(DETC)2NO ratio allows not only to better assess the intensity and location of NO production, but also provides information about the parameters of the tumor structure. Unfortunately, in this way, the assessment of NO can only be performed in tumors growing in animals, because exogenous NO spin trapslike DETC, are toxic and are not used in patients, therefore, for diagnostic purposes in humans, we can only assess the level of HbNO in the tumor. However, in tumors growing in an animal model, the HbNO/Fe(DETC)2NO ratio can be used for research on the production of NO and the structure of tumors. The low HbNO/Fe(DETC)2NO ratio (balance strongly shifted towards NO and DETC complexes) indicates intensive NO production, poor vascularization and lack of areas of hemorrhagic necrosis in these tumors (e.g. Ehrlich carcinoma). The high HbNO/Fe(DETC)2NO ratio in tumors (the balance strongly shifted towards NO-Hb complexes) indicates intensive NO production within the hemorrhagic, very good vascularization of these tumors, and strong extravasation of blood (e.g. S91 melanoma). The HbNO/ Fe(DETC)2NO ratio close to 1, with simultaneously strong HbNO and Fe(DETC)2NO signals, indicates high NO production in these tumors, good vascularization, and high heterogeneity in terms of blood extravasation into the tissue and the formation of foci of hemorrhagic necrosis (e.g. L5178Y lymphoma).

I recommend to provide EPR spectra corresponding to different tumor samples as well as images mentioned in Section "Materials and Methods" in Supplementary information.

The spectra show not only the aspect of NO itself, but also image and carry information about the simultaneous presence of various complexes in tumors (triplet signal of 5-coordinated HbNO complexes, broad signal of 6-coordinated HbNO complexes, Fe(RS)2(NO)2 signal, Fe(DETC)2NO, Cu(DETC)2 signal, free radical signal).

We do not want to transfer representative EPR spectra of tumors (Results) or tumor spectra (Methods) showing the signal amplitude measurement. Especially since one reviewer suggests moving Figure 6 with complexes from methods to the beginning of the results for a better understanding of the results, which is worth considering.

Also, the origin of the samples which EPR spectra are presented at Figure 6 should be specified.

We have added in the description of Figure 6: In A. Native melanoma S91. B. Native Lymphoma L5178Y. C. Lymphoma L5178Y with exogenous DETC.

I believe that manuscript by D. Szczygiel and co-authors can be published in Special Issue "Nitric Oxide Biosynthesis Pathway and Nitric Oxide Signaling in Human Diseases" of IJMS after appropriate improvement: practical value of the method described or perspectives of its practical application must be discussed in the text; title of Fig. 6 should be clarified; Supplementary materials (images, EPR spectra) should be provided.

Comments on the Quality of English Language

Line 54: "Exogenous" should be instead of "egzogenous"

Corrected

Line 436: "Biotin" should be instead of "Biotyn"

Corrected

Reviewer 2 Report

Comments and Suggestions for Authors

Dear Authors, see comments below.

1) The text of the article and the figures do not provide an adequate description of the EPR spectra along with simulation (for example, the EASYSPIN program). Due to the absence of a spin Hamiltonian, the identification of signals remains questionable.

2) The authors compared amplitudes determined from peak to peak (Ip-p). Spectral simulations will allow you to establish the integral intensity, which is more true and reliable for comparison.

3) In my opinion, Figure 2 could be replaced with a histogram for more convenient visualization.

In current case, it is perceived as some incomprehensible function F(x), with an undesignated ordinate axis.

4) Fig.3. A. The exponential curve simulation data in Figure 3a is extremely questionable. Firstly, the scatter is too large, and those two points on the right can only have a statistical deviation. Secondly, the intensity of the EPR spectrum is an extremely sensitive quantity, depending, for example, on the quality factor of the resonator, where Ip-p is proportional to Q. 

Thus, the value of the confidence interval does not allow us to judge the correct behavior (trajectory) of the theoretical curve, which requires more accurate experimental measurements.

5) The authors state that all ESR measurements were obtained with a modulation amplitude of 10 G. The authors do not say a word about line width, which means that inhomogeneous broadening may occur due to the overmodulation effect and, as a consequence, distortion of the line and intensity (amplitude) of the EPR spectrum.

Last comment. 

All your references in all chapters of the article are very old in years. There are practically no publications (references) over the last 5-7 years, and literally only a few since 2005. I am not against old classical experimental and theoretical works, but the relevance of the manuscript is confirmed by the references to the release of the last 5 years. As a result, the authors did not study fresh and relevant scientific works, at least on related studies.

The nature of the presented experimental data and the corresponding comparative analysis require strong revision.

Author Response

Dear Authors, see comments below.

1) The text of the article and the figures do not provide an adequate description of the EPR spectra along with simulation (for example, the EASYSPIN program). Due to the absence of a spin Hamiltonian, the identification of signals remains questionable.

Simulations of EPR spectra of various types of HbNO complexes are presented in this newely cited paper: Dutka et al. 2019 (a chapter).

2) The authors compared amplitudes determined from peak to peak (Ip-p). Spectral simulations will allow you to establish the integral intensity, which is more true and reliable for comparison.

The reviewer is perfectly convincing. But, in native mouse tumors, EPR spectra are derived from the composite signals of many different paramagnetic centers: signals of 5-coordinated Fe-NO complexes α-chains of Hb, 5-coordinated Fe-NO complexes β-chains of Hb, non-heme Fe(RS)2(NO)2 complexes, and free radical centers are present. When a DETC trap is administered, signals of Fe(DETC)2NO complexes and Cu(DETC)2 complexes also appear on the EPR spectrum. In our opinion, quantitatively separating all these signals from each other in the recorded EPR spectrum and determining the integral intensity of each of the recorded signals is burdened with high uncertainty and will depend on the method of conducting the simulation and the complexity of the measured spectrum. In a situation where the spectrum consists of only a single signal, measuring the integral intensity is the most reliable, but as the number of superimposed signals increases, measuring the integral intensity of each of them will be burdened with an increasing error. Therefore, we believe that for comparisons it will be more reliable to measure the amplitude of the selected line of the HbNO signal (5-coordinated complexes Fe-NO β-chains of Hb) and the Fe(DETC)2NO signal in the situation of superimposing many signals.

3) In my opinion, Figure 2 could be replaced with a histogram for more convenient visualization. In current case, it is perceived as some incomprehensible function F(x), with an undesignated ordinate axis.

In some papers such illustration served to depict the spread and the mean value compared between different tumor lines. The histogram may have lost some important information

4) Fig.3. A. The exponential curve simulation data in Figure 3a is extremely questionable. Firstly, the scatter is too large, and those two points on the right can only have a statistical deviation. Secondly, the intensity of the EPR spectrum is an extremely sensitive quantity, depending, for example, on the quality factor of the resonator, where Ip-p is proportional to Q. Thus, the value of the confidence interval does not allow us to judge the correct behavior (trajectory) of the theoretical curve, which requires more accurate experimental measurements.

We agree that more data would be useful to make the fitted curve shape in Fig. 3A more reliable. However, it can be seen that in Fig. 3B, for the same type of tumors (lymphoma L5178Y), on the same day of growth (day 8), we have a very similar shape of the fitted curve. The different host of this tumor only causes the HbNO and Fe(DETC)2NO signals to be stronger in the allogeneic Swiss host (large difference in tissue antigens) than in the syngeneic host (small difference in tissue antigens), but the shape of the correlation of the HbNO and Fe(DETC)2NO signals is very close.

We have supplemented the paper with such, additional data, which, however, lead to a new publication, and here we decided to cite them only as “unpublished results”.

5) The authors state that all ESR measurements were obtained with a modulation amplitude of 10 G. The authors do not say a word about line width, which means that inhomogeneous broadening may occur due to the overmodulation effect and, as a consequence, distortion of the line and intensity (amplitude) of the EPR spectrum.

The modulation amplitude of 10 Gs was checked many times in the past in other experiments, already published, and chosen because in biological samples the signals of HbNO and Fe(DETC)2NO complexes are often quite weak and at a lower modulation amplitude the weak signals are beyond detection. We are aware that using a modulation amplitude of 10 Gs may lead to overmodulation, which may widen and distort the signal lines. However, all measurements were performed with a modulation amplitude of 10 Gs and it can be expected that if there is a broadening of a given type of signal, this broadening is very similar in each case and does not significantly affect the comparison of signals between groups. However, we found that for a better assessment of tumors, it is better to obtain even slightly broadened signals in the same way in each group than to completely lose information about the amplitude of EPR signals in many tumor samples.

Last comment. 

All your references in all chapters of the article are very old in years. There are practically no publications (references) over the last 5-7 years, and literally only a few since 2005. I am not against old classical experimental and theoretical works, but the relevance of the manuscript is confirmed by the references to the release of the last 5 years. As a result, the authors did not study fresh and relevant scientific works, at least on related studies.

We have supplemented the manuscript with some newer papers and chapters. However, in some cases (e.g. NO in L5178Y lymphoma) the topic was not taken into consideration for recent time.

The nature of the presented experimental data and the corresponding comparative analysis require strong revision.

Reviewer 3 Report

Comments and Suggestions for Authors

In this manuscript, the authors investigated the levels of nitric oxide (NO) in different tumor tissues using both exogenous ferrous diethyldithiocarbamate (Fe(DETC)2NO) and endogenous hemoglobin (HbNO) spin traps through electron paramagnetic resonance (EPR). While the results provide valuable insights, the manuscript suffers from poor writing and organization. I recommend major revisions before publication, and below are specific points to address:

1.      There is a lack of connection between sentences. For example,

a)        The transition from the first to the second sentence in the abstract is unreasonable.

b)        The consequence of "the accessibility of Hb as a spin trap for NO" is unclear in lines 44-45.

c)        The sentence starting with "but" in lines 67-70 does not flow logically from the preceding sentences.

2.      Introduction:

a)        Introduce the rationale for using EPR over other technologies.

b)        Consider providing the chemical structures of key compounds, such as HbNO, Fe(DETC)2NO, and nitrite.

3.      Results

a)        Provide clearer explanations and improve the flow between paragraphs.

b)        Integrate some of the content from the Discussion section into the Results to enhance readability.

4.      Fig 1 and Fig 2

a)        The description of Fig 1 and Fig 2 is disorganized. Consider reorganizing the figures or the language in the main text for clarity.

b)        Avoid jumping back and forth between figures to prevent confusion.

5.      Table 1 and unshown data

a)        Consider incorporating tumor parameters from Table 1 and unshown data from lines 126 & 130 into a figure.

b)        Explain the rationale for how the Host and day of tumor growth were chosen.

6.      Fig 4

a)        Explain the rationale for choosing the combination of HbNO-S91 and Fe(DETC)2NO-L5178Y.

b)        Correct the title of the right bar chart in Fig 4B to Fe(DETC)2NO as described in the caption.

c)        Check the data in Fig 4B, as the bar height seems not consistent with the provided EPR curves.

7.      Move Fig 6 to an earlier position when the EPR data first appear in the results. Explain how the amplitude was determined from the curves.

8.      Remove background information not derived from Conclusion, such as the three factors influencing the appearance of EPR-detectable HbNO complexes in tumor tissues in vivo.

Comments on the Quality of English Language

1.      There is a lack of connection between sentences. For example,

a)        The transition from the first to the second sentence in the abstract is unreasonable.

b)        The consequence of "the accessibility of Hb as a spin trap for NO" is unclear in lines 44-45.

The sentence starting with "but" in lines 67-70 does not flow logically from the preceding sentences.

Round 2

Reviewer 1 Report

Comments and Suggestions for Authors

Authors made appropriate corrections in the manuscript and replied all the questions addressed. I believe that the manuscript can be published in its present form.

Author Response

We are again very grateful for thorough and painstaking review which improved so much the text.

Reviewer 2 Report

Comments and Suggestions for Authors

I am quite happy with the answers to my questions. Revision's article looks pretty good and may be accepted.

Author Response

Thank you very much for your work to improve on the manuscript. We are really grateful.

Reviewer 3 Report

Comments and Suggestions for Authors

After the authors’ modifications, the manuscript is clearer. However, it still suffers from poor organization and description, which make it unsuitable for publication in the International Journal of Molecular Sciences. Here are my key points:

1.      The flow from fig1-fig3-fig1-table1-fig1-table1-fig2-fig3 lacks logical connection and appears disorganized. While I understand the authors' attempt to demonstrate the significance of tumor vascularization, different host types (DBA/2 and Swiss), and hemorrhagic necrosis in HbNO formation, presenting these factors together in Figure 1 and Table 1 is confusing. I suggest segregating these factors into separate figures and introducing them as needed for clarity.

2.      In the updated Figure 5B, the detection of HbNO rather than Fe(DETC)2NO after the addition of DETC is confusing. Furthermore, the absence of data deviation raises concerns about the reliability of the EPR signals.

Author Response

Dear Reviewer 3, your thorough re-revision improved even more on the precision and clearity of the manuscript (if possible) Below the details accordingly.

  1. The flow from fig1-fig3-fig1-table1-fig1-table1-fig2-fig3 lacks logical connection and appears disorganized.

We have changed the order of the figures in Results chapter to improve reading fluency. Now the order is:

Figure 1. Qualitative assessment of NO complexes (EPR spectra) and vascularization (histological images with immunohistochemical staining of vessels with CD31 and macro images of blood supply) in various types of tumors.

Figure 2. Quantification of NO complexes (heme HbNO and non-heme Fe(DETC)2NO) in various types of cancer.

Table1. Quantitative assessment of vascularization parameters (MVD, TVA, MAM) in various types of cancer.

Figure 3. Quantitative correlation of EPR signals of heme HbNO complexes with the total vascular area (TVA) in various types of cancer.

Figure 4. Quantitative correlation of the levels of heme HbNO complexes with non-heme Fe(DETC)2NO complexes in various types of cancer.

Figure 5. Effect of the development of hemorrhagic necrosis on the HbNO complexes level in the tumor necrosis, in the living vascularized tumor tissue and in the peripheral blood of the tumor host.

Figure 6. The influence of blood extravasation into the tumor tissue, during the mechanical process of the blood vessels destruction, on the level of HbNO complexes formed in the tumor.

While I understand the authors' attempt to demonstrate the significance of tumor vascularization, different host types (DBA/2 and Swiss), and hemorrhagic necrosis in HbNO formation, presenting these factors together in Figure 1 and Table 1 is confusing. I suggest segregating these factors into separate figures and introducing them as needed for clarity.

Figure 1 shows histological images of vascularization and macro images of the degree of blood supply of various types of tumors (L5178Y lymphoma, Ehrlich carcinoma, S91 melanoma) growing in the host appropriate for a given type of tumor (DBA/2 mice for L5178Y and S91 tumors, Swiss mice for EC tumors), together with representative spectra of EPR signals of NO complexes in these tumors. For S91 melanoma, images are presented separately for the area of the cortical layer and separately for the area of central hemorrhagic necrosis, which together constitute the entire S91 tumor. We do not analyze different hosts for the same cancer type in Figure 1. We think that the current set of results in Figure 1 should be understandable. Breaking Figure 1 into several subsequent Figures would also probably result in too many Figures in the entire work (currently there are 7 Figures).

Table 1 summarizes the quantitative assessment of histological images of the tumors vascularization presented in Figure 1. Three parameters of vascularization (MVD, TVA, MAM) of various types of tumors (L5178Y lymphoma, Ehrlich carcinoma, S91 melanoma) growing in a host appropriate for a given type of tumor (DBA/ 2 mice for L5178Y and S91 tumors, Swiss mice for EC tumors) were assessed. Vascularization parameters in S91 tumors were assessed only in the cortical layer of the tumors, because in the hemorrhagic necrosis the vessels are already destroyed and their assessment is not possible. L5178Y lymphoma tumors have quite aggressive growth and are able to grow not only in syngeneic DBA/2 mice, but also in an atypical hosts such as allogeneic Swiss mice. Therefore, additionally for L5178Y lymphoma tumors in Swiss mice, the values of vascularization parameters are also included. However, it seems to us that creating a separate table/figure with vascular data to compare L5178Y tumors in two different hosts (DBA/2 and Swiss) is probably not entirely justified. An in-depth analysis of the host's influence on vascularization parameters and the level of NO complexes in the tumor is not the aim of this study. What is more interesting is the simultaneous comparison of L5178Y and EC tumors growing in the same host (Swiss mice) in one table and showing that the type of tumor has a key impact on the development of blood vessels, and the influence of the type of host on the development of tumor vascularization is significantly smaller. Perhaps the following reorganization of Table 1 will make it clearer:

Table 1. Parameters of solid tumor vascularization in L5178Y, EC and S91 tumors. MVD – microvessel density (number of vessels/mm2); TVA - total vascular area expressed in mm2 of vessel area within 1 mm2 of tumor tissue section (percentage of area occupied by vessels [%]); MAM – mean area of microvessels (vessels size [μm2] calculated as TVA/MVD). Means were obtained from 3 histological slices (L5178Y lymphoma, Ehrlich carcinoma, spleen) and 9 histological slices (Cloudman S91 melanoma). All results are shown as mean ± SD.

Tumor growth parameters

Tumor vascularization parameters

Solid tumors

Host

Time [days]

TVA

MVD

MAM

Ehrlich carcinoma

Swiss

18

0.6± 0.7

23 ± 29

230±121

L5178Y lymphoma

DBA/2

5

3.3± 0.6

185 ± 25

176±71

Swiss

5

4.9± 0.6

265 ± 11

190±112

DBA/2

8

5.4±0.6

354 ± 26

150±97

S91 melanoma

DBA/2

27

8.0±2.6

199±73

490±143

Spleen (control)

DBA/2

-

2.8± 0.1

841 ± 120

40±18

In Figure 1 and Figure 2 for S91 tumors, there is a separate assessment of NO complexes in the superficial layer of tumors and separately in the central hemorrhagic necrosis of tumors in order to have a comprehensive picture of NO complexes in these tumors, when compared with other types of tumors ( LY, EC). Additionally, results for LY tumors were presented both for DBA/2 and Swiss host. Firstly, to assess the impact of the host on the NO complexes and vascular parameters in the tumor. Secondly, to be able to compare different types of tumors growing in the same type of host (LY and S91 in DBA/2 mice - inbred strain, and LY and EC in Swiss mice - outbred strain), and thus be able to compare the influence of the tumor type and the host type on measured parameters.

  1. In the updated Figure 5B, the detection of HbNO rather than Fe(DETC)2NO after the addition of DETC is confusing. Furthermore, the absence of data deviation raises concerns about the reliability of the EPR signals.

The purpose of Figure 5 is to show that the presence of strong HbNO signals in tumors is caused by the development of areas of hemorrhagic necrosis in the tumor tissue, and that areas of hemorrhagic necrosis are the main site of accumulation of HbNO complexes in the tumor tissue. Therefore, only HbNO complexes were analyzed in Figure 5.

In advanced S91 melanoma tumors (day 27), central hemorrhagic necrosis always develops, and therefore the S91 tumors are a very good, repeatable research model of hemorrhagic necrosis. Hence, Figure 5A shows results from a large number of S91 tumors with visible hemorrhagic necrosis (N=12), and the results obtained have statistical significance at the p<0.001 level.

In advanced L5178Y lymphoma tumors (day 11), visible central hemorrhagic necrosis develops very rarely. Unfortunately, the longer development time of the L5178Y tumor usually leads to a significant deterioration in the condition of the mice and the need to end the experiment, without the possibility of assessing the tumors after a longer growth time. During the studies with L5178Y tumors, only a one tumor with visible central hemorrhagic necrosis was obtained, which grew in Swiss mice injected with DETC. Results obtained for samples prepared from this L5178Y tumor clearly match the picture observed in S91 tumors, so we decided to show the measured HbNO signals from samples of this tumor in Figure 5B. We did not include the values of dispersion between tumors in the case of analysis for a single tumor. For the same reason, we did not include the quantification of Fe(DETC)2NO complex signals in Figure 5B due to the lack of more cases of tumors with hemorrhagic necrosis that would also show the presence of Fe(DETC)2NO complexes. We are currently conducting preliminary studies in other tumor models that sometimes show the presence of hemorrhagic necrosis and signals of Fe(DETC)2NO complexes (data not shown). These results are quite consistent with the results presented here for this single tumor L5178Y (strong Fe(DETC)2NO signals apart from hemorrhagic necrosis and weak Fe(DETC)2NO signals in hemorrhagic necrosis), however, the analysis of Fe(DETC)2NO and HbNO complexes in necrotizing tumors (in the context of immune activation)  will be the subject of a separate publication.

Round 3

Reviewer 3 Report

Comments and Suggestions for Authors

Thanks for the authors explanition. the manuscript is now clear to me. I'm pleased that it can be accepted in its current form.